# Visual Integration of Genome-Wide Association Studies and Differential Expression Results with the Hidecan R Package

**DOI:** 10.3390/genes15101244

**Published:** 2024-09-25

**Authors:** Olivia Angelin-Bonnet, Matthieu Vignes, Patrick J. Biggs, Samantha Baldwin, Susan Thomson

**Affiliations:** 1The New Zealand Institute for Plant and Food Research Limited, Palmerston North 4442, New Zealand; 2School of Mathematical and Computational Sciences, Massey University, Palmerston North 4442, New Zealand; m.vignes@massey.ac.nz; 3School of Food Technology and Natural Sciences, Massey University, Palmerston North 4442, New Zealand; p.biggs@massey.ac.nz; 4School of Veterinary Science, Massey University, Palmerston North 4442, New Zealand; 5The New Zealand Institute for Plant and Food Research Limited, Christchurch 8140, New Zealand; samantha.baldwin@plantandfood.co.nz (S.B.); susan.thomson@plantandfood.co.nz (S.T.)

**Keywords:** visualisation, genome-wide association studies, differential expression, autotetraploid, R package

## Abstract

Background/Objectives: We present hidecan, an R package for generating visualisations that summarise the results of one or more genome-wide association studies (GWAS) and differential expression analyses, as well as manually curated candidate genes, e.g., extracted from the literature. This tool is applicable to all ploidy levels; we notably provide functionalities to facilitate the visualisation of GWAS results obtained for autotetraploid organisms with the GWASpoly package. Results: We illustrate the capabilities of hidecan with examples from two autotetraploid potato datasets. Conclusions: The hidecan package is implemented in R and is publicly available on the CRAN repository and on GitHub. A description of the package, as well as a detailed tutorial, is made available alongside the package. It is also part of the VIEWpoly tool for the visualisation and exploration of results from polyploids computational tools.

## 1. Introduction

Genome-wide association studies (GWAS) enable researchers to investigate genomic regions associated with a trait of interest [1]. Often, the ultimate goal of such analyses is to highlight causal genes that are involved in biological processes relating to the phenotype under study; for example, to detect disease-related genes that can be targeted by new drugs, or to facilitate the selection of desirable characteristics in breeding programmes. Therefore, association studies can be complemented with the acquisition of transcriptomics data [2,3,4]. These can be used to detect genes whose expression is associated with the trait under study through a differential expression (DE) analysis. The results from both GWAS and DE are compared in terms of an overlap between the genomic regions containing significant markers and differentially expressed genes. Results from both analyses are also typically compared with previous results from the literature in order to assess which findings are substantiated by previous research and which are novel results that must be confirmed with follow-up studies.

GWAS results are usually displayed through a Manhattan plot, which represents the score of each genomic variant (i.e., −log10 of its *p*-value) against its physical position along the genome [5]. DE results are instead typically represented with a volcano plot, which displays the DE score of the genes against their log2-fold change [6]. There are two main drawbacks to using these plots. First, they cannot be easily supplemented with manual information: for example, genes of interest from the literature. Second, Manhattan plots and volcano plots cannot be aligned and combined to represent both GWAS and DE results in a single figure. Therefore, the overlap between GWAS and DE results is often presented in tables (e.g., [3]) or Venn diagrams (e.g., [4]). In recent years, the Circos plot has emerged as an attractive design to display information related to genomics data [7] and thus can be used to visualise the results of both GWAS and DE. It is notably possible to combine several layers of information by stacking several circles on top of each other in a single Circos plot [3,8]. However, the circular layout of Circos plots makes it difficult to precisely compare the different layers and, thus, to identify genomic regions of interest in which the information from several layers overlaps.

In this paper, we present the hidecan R package, which can be used to generate so-called HIDECAN plots [9] that integrate GWAS and DE results as well as candidate genes (from the literature) into one graphic. The package is publicly available on the CRAN repository (https://CRAN.R-project.org/package=hidecan, accessed on 22 September 2024) and on GitHub (https://github.com/PlantandFoodResearch/hidecan, accessed on 22 September 2024) with a detailed tutorial available at https://plantandfoodresearch.github.io/hidecan/ (accessed on 22 September 2024). Here, we showcase how the package can be used to generate such plots with little code. We also demonstrate how GWAS results from autotetraploid organisms obtained with the R package GWASpoly (https://github.com/jendelman/GWASpoly, accessed on 22 September 2024) can be displayed with hidecan. Lastly, we show an example of adding other data types to a HIDECAN plot.

## 2. Materials and Methods

### 2.1. Input Data

The main function of the hidecan package is the hidecan_plot() function. It takes as input data-frames of GWAS results, DE results, and candidate genes. These data-frames should contain information about the genomic position of the genomic variants and genes (i.e., chromosome and physical position or start and end), as well as GWAS or DE scores and log2-fold change. In particular, a number of mandatory columns are expected in the input data-frames:*chromosome* (for GWAS, DE, and candidate gene tables): column giving the ID of the chromosome on which each marker or gene is located;*position* (for GWAS tables): the physical location in the base pair (bp) of the markers along the chromosome;*start* and *end* (for DE and candidate gene tables): the start and end positions (in bp) of the genes along the chromosome. These two columns are used by hidecan to calculate the position of the genes on the plot as the half-way point between their start and end positions;Either *score* or *padj* (for GWAS and DE tables): either the score (i.e., −log_10_(*p*-value) or the adjusted *p*-value of the markers or genes in the GWAS or DE results. If a *score* column is provided, the *padj* column will be ignored. If only a *padj* column is provided, it will be used to compute the score of the markers or genes;Either *foldChange* or *log2FoldChange* (for DE tables): either the fold-change or the $\log_{2}$(fold-change) of the genes in the DE results. If a *log2FoldChange* column is provided, the *foldChange* column will be ignored. If only a *foldChange* column is provided, it will be used to compute the $\log_{2}$(fold-change) of the genes;*name* (for candidate genes tables): the name of the genes that will be displayed in the HIDECAN plot. These can be set to NA (missing values in R) for any of the genes to remove their label in the resulting plot.

Note that any number of other columns might be present in the input data-frames; they will be ignored by the package’s functions.

The user needs to set a significance threshold on the GWAS scores, DE scores, and DE $\log_{2}$(fold-change) in order to select genomics variants and genes that will be displayed in the HIDECAN plot (Figure 1A–D). These are passed as arguments to the hidecan_plot() function; only markers and genes with a score and $\log_{2}$(fold-change) equal to or higher than the set thresholds will be shown in the resulting plot. This filtering is performed internally through the apply_threshold() function. Markers and genes with missing scores or $\log_{2}$(fold-change) are automatically excluded from the plot.

### 2.2. Additional Parameters

Several other parameters included in the hidecan_plot() function allow the user to control different aspects of the plot, such as which chromosomes should be displayed or the size of the points in the graphic. In particular, by default, the length (in bp) of each chromosome is calculated internally as the maximum marker or gene position across the input data-frames through the compute_chrom_length() and combine_chrom_length() functions. The user can instead provide values for the length of each chromosome, which is useful if there are no markers or genes towards the end of the chromosomes, for example. By default, entire chromosomes are shown in the HIDECAN plots. Users can instead specify for one, some, or all chromosomes the minimum and maximum positions to show. This effectively allows the user to zoom in on specific regions of interest.

### 2.3. Output

The hidecan_plot() function returns a HIDECAN plot constructed with the ggplot2 [10], ggrepel [11] and ggnewscale [12] packages. In a HIDECAN plot, the *x*-axis represents the physical location (in bp) along the chromosomes. The different chromosomes are shown in separate facets (boxes) in the plot. The positions of markers and genes meeting the significance thresholds set by the user are represented as points, which are split along the *y*-axis according to their type: peaks in the GWAS results, differentially expressed genes, or candidate genes. Vertical lines are used to facilitate the comparison of their positions across the *y*-axis tracks. Points representing significant markers and genes are coloured according to their GWAS or DE score; in the case of genes, there is the option to colour them according to their $\log_{2}$(fold-change) instead. For candidate genes, a label depicting the gene name is connected to each point.

### 2.4. Custom Data Types

While the original goal of HIDECAN plots is to visualise GWAS and DE results, in principle, any data type that can be mapped to a physical position on the genome can be included in these plots. This includes, for example, results from QTL mapping, differentially methylated regions, or chromatin accessibility data. In order to accommodate for this possibility, the hidecan_plot() function also accepts custom data-frames of genomic features. These data-frames must contain, at the minimum, the following columns: *chromosome*, *position*, and *score*, as defined for GWAS or DE results. In the case of genomic regions that span multiple bp, the user must calculate the position of the corresponding point on the plot. The user can then customise the name given to the new data tracks, choose a colour palette to display the score for this data type, or show labels for the genomic features of interest. This provides a flexible approach where users can come up with their own integrative plots without relying on the package implementing each specific data type.

### 2.5. Shiny Application

Alternatively to using the package’s functions, a Shiny app is made available, which is launched via the run_hidecan_shiny() function. The app enables the construction of a HIDECAN plot via a visual interface rather than with code. The GWAS results, DE results, and candidate genes lists are uploaded to the app as comma-separated (csv) files. Note that the same requirements regarding mandatory columns for each type of data input apply. Sliders are used to set the score and fold-change thresholds, and additional plot annotations (such as title or legend position) can be set. The generated HIDECAN plot can then be saved as either a PNG or PDF file. Currently the Shiny app does not accept other custom data types; this is only possible by using the package functions directly.

### 2.6. Conversion from GWASpoly Package

Often, the GWAS and DE analyses are performed using popular R packages such as edgeR [13] or DESeq2 [14]. For example, GWASpoly [15] is an R package for performing GWAS for autotetraploid organisms. Through the GWASpoly package, it is possible to obtain GWAS scores for several traits at once and with different genetic models. The package then offers several options to compute a significance threshold on the genomic variants score for each trait and genetic model. In order to facilitate the construction of HIDECAN plots from GWASpoly results, we offer a helper function to extract the necessary information directly from the output of the GWASpoly package. The hidecan_plot_from_ gwaspoly() function accepts as input a GWAS result object obtained from GWASpoly and extracts the variant scores computed for each analysed trait and each genetic model tested, as well the significance thresholds calculated by GWASpoly for each trait and genetic model. These are used to produce a HIDECAN plot, which, by default, displays the GWAS results obtained for all traits and genetic models. The user has the option to select which traits and genetic models should be shown. Such helper functions reduce the amount of data wrangling that users must perform to convert the output of these packages into a suitable format for the hidecan package. Due to the modular construction of the package, it is possible to easily implement additional helper functions for other diploid- or polyploid-focused packages if the need arises.

### 2.7. Availability through VIEWpoly

The hidecan package has been included in the visualisation tool VIEWpoly [16]. VIEWpoly is an R package and Shiny app for visualising and integrating results from polyploid genetic and genomic analyses tools. It provides an intuitive interface that facilitates the interactive exploration of linkage mapping and Quantitative Trait Loci (QTL) analysis results obtained from a suite of packages such as MAPpoly [17] and polyqtlR [18]. Users can explore the QTL results and linkage maps in more detail or use the implemented genome browser to explore the relationships between QTL regions and a corresponding genome annotation. In the newest version of the package (v0.4.1), it is possible to upload GWAS and DE results as well as candidate gene lists as csv files, or GWASpoly results as RData files, into the VIEWpoly Shiny app. A HIDECAN plot is generated from these input files using the hidecan package. Users can further explore and customise the HIDECAN plot and save the resulting figure as a file with one of several supported extensions.

## 3. Results

### 3.1. Example for Tuber Bruising in the Autotetraploid Potato

We demonstrate the use of the hidecan package with a dataset from a breeding population of tetraploid potatoes, originally published in [9]. A GWAS analysis was applied to 72,847 genomic variants obtained from 158 progeny plants from a half-sibling breeding population to identify variants associated with differences in tuber bruising intensity response. The results were compared with candidate genes from previous GWAS or QTL mapping studies on potato tuber bruising. In addition, a DE analysis was used to compare the expression of 25,163 transcribed genes between low- and high-bruising tubers two hours after bruising. From the literature, a list of 42 candidate genes found previously associated with tuber bruising was assembled.

A subset of these results is made available in the hidecan package through the get_example_data() function. From the complete GWAS results table, half of the genomic variants with a GWAS score < 3.5 were randomly selected and consequently discarded, yielding a dataset with GWAS scores for 35,481 variants. Similarly, half of the transcribed genes in the DE results table with an adjusted *p*-value > 0.05 were randomly selected and discarded, yielding a dataset with DE results for 10,671 transcribed genes. This filtering was performed to reduce the size of the datasets (in accordance with CRAN policies) but ensures that all significant markers and genes are retained in the datasets. Finally, some of the candidate genes located on chromosome 3 were removed from the example dataset for better clarity in the resulting HIDECAN plot, leaving 32 candidate genes.

We can visualise this dataset as a HIDECAN plot. For this example, we set the significance threshold for the GWAS analysis to 4 (corresponding to a *p*-value of $1\times 10^{-4}$). We use for the DE results a significance threshold of 1.3 (corresponding to an adjusted *p*-value of 0.05) and a $\log_{2}$(fold-change) threshold of 0 (which amounts to no filtering based on the $\log_{2}$(fold-change)). These thresholds are set through the score_thr_gwas, score_thr_de, and log2fc_thr arguments, respectively:


library(hidecan)

data <- get_example_data()

hidecan_plot(
    gwas_list = data[["GWAS"]], # data-frame of GWAS results
    de_list = data[["DE"]],     # data-frame of DE results
    can_list = data[["CAN"]],   # data-frame of candidate genes
    score_thr_gwas = 4,         # sign. threshold for GWAS
    score_thr_de = 1.3,         # sign. threshold for DE
    log2fc_thr = 0              # log2-fold change threshold for DE
)


Figure 1E showcases the resulting HIDECAN plot for chromosomes 0 to 5. The full HIDECAN plot, as well as the code used to generate the figure and details about the dataset, are presented in Appendix A.

### 3.2. Visualising Output from the GWASpoly Package

We showcase how the hidecan package can be used to visualise GWAS results from autotetraploid organisms obtained with the GWASpoly package. We used the example dataset from the GWASpoly publication [15], in which genomic information and phenotype measurements were obtained for 221 tetraploid potato lines from the SolCAP diversity panel. We focused on the analysis of three traits, namely tuber eye depth, tuber shape, and sucrose, and we performed a GWAS analysis with four different genetic models, namely general, additive, reference simplex dominant (1-dom-ref), and alternate simplex dominant (1-dom-alt). The code used to perform the analysis with GWASpoly is shown in Appendix A. The GWASpoly output for this example dataset is available through the hidecan package and can be directly displayed in a HIDECAN plot through the following code:


library(hidecan)
library(GWASpoly)

# Reading the example GWASpoly output
gwaspoly_example_file <- system.file(
    "extdata/gwaspoly_res_thr.rda",
    package = "hidecan"
)
gwaspoly_res_thr <- readRDS(gwaspoly_example_file)

hidecan_plot_from_gwaspoly(
    gwaspoly_res_thr,            # GWASpoly output
    remove_empty_chrom = TRUE,   # only display chrom with sign. markers
    chrom_limits = list(         # zoom in on specific genomic regions


	"5" = c(2e6, 2.5e6),     # (limits in bp)
        "8" = c(53e6, 53.5e6),
        "10" = c(48.5e6, 49e6),
        "11" = c(0.5e6, 1e6)
    )
)


The resulting plot (Figure 2) facilitates the comparison of the results from the different genetic models. In this case, we can easily see that one marker is found significantly associated with tuber eye depth with both the general and additive models, while two markers are only found significant with the additive model. Conversely, none of the markers reach the significance threshold for this trait with either of the simplex dominant models.

The HIDECAN plot also allows us to compare GWAS results across several traits of interest. This is especially interesting when studying related or correlated phenotypes, as it can allow us to identify shared causative variants and genes influencing several traits at once [19,20]. In this example, we can observe that the two markers found significantly associated with tuber eye depth on chromosome 10 are also found associated with tuber shape. On the other hand, the significant marker on chromosome 5 is specific to tuber eye depth. None of these markers are found significantly associated with sucrose content. We note that the ability to compare results from different traits is not restricted to GWAS results from GWASpoly; users can provide more than one table of GWAS and/or DE results to the hidecan_plot() function as well.

### 3.3. Adding Custom Data Tracks

Finally, we illustrate the ability of the hidecan package to accommodate data other than GWAS and DE results. In Figure 3, we show a simulated example where additional data types, specifically results from a QTL mapping analysis and differentially methylated regions (DMRs), were added to a HIDECAN plot containing GWAS results. The code used to construct the plot with these custom data types is shown below; it involves choosing a colour palette to represent the score of the differentially methylated regions and specifying the name of the tracks for both data types, the colour of the vertical lines that show the position of significant features, as well as whether the name of these features should be displayed (in this case, we show the QTL region labels). The code used to generate this simulated dataset is given in Appendix A.


library(hidecan)
library(ggplot2)

# Setting up a colour palette for differential methylation score
dmr_palette <- scale_fill_viridis_c(
  option = "rocket",
  name = "DM score",
  guide = guide_colourbar(
    title.position = "top",
    title.hjust = 0.5
  )
)

# Aesthetics for the custom data types
plot_aes <- list(
  QTL_data = list(
    y_label = "QTL regions",
    line_colour = "darkgoldenrod2",
    point_shape = 18,
    show_name = TRUE,
    fill_scale = NULL
  ),

  DMR_data = list(
    y_label = "Diff. methylated regions",
    line_colour = "orchid",
    point_shape = 24,
    show_name = FALSE,
    fill_scale = dmr_palette
  )
)

# Specify the data type of each custom input table
attr(sim_data[["QTL"]], "aes_type") <- "QTL_data"
attr(sim_data[["DMR"]], "aes_type") <- "DMR_data"

hidecan_plot(
  gwas_list = sim_data[["GWAS"]],
  score_thr_gwas = 2,
  custom_list = list(sim_data[["QTL"]], sim_data[["DMR"]]),
  score_thr_custom = 2,
  custom_aes = plot_aes
)


While the code required to generate a HIDECAN plot with custom tracks is a bit more complex, it provides a lot of flexibility for users, as they can add any data type desired to the plot.

## 4. Discussion

We present the hidecan R package for generating HIDECAN plots that display genomic variants and genes highlighted by GWAS and DE analyses, alongside manually curated candidate genes of interest. The HIDECAN plot provides in a single visualisation a genome-wide overview of these three types of information. Critically, it allows users to easily identify genomic regions in which association with a trait of interest is supported at both the genomics and transcriptomics levels, and/or supported by previous findings. It also facilitates the comparison of GWAS and/or DE results across multiple traits or multiple models or methods. The hidecan R package provides simple functions that allow the user to rapidly generate such HIDECAN plots from tables of GWAS and DE results. It also offers a Shiny app that enables the construction of plots through an interactive interface rather than through code. Finally, it provides helper functions that extract results from the output of packages performing the GWAS or DE analysis to facilitate the use of the package in an R analysis pipeline. We note that while the original goal of HIDECAN plots was to integrate GWAS and DE results, any data that can be mapped to a physical location on the genome can be included in a HIDECAN plot, with the user controlling the look of the tracks for these other data types.

One strength of the hidecan package is that it provides a genome-wide overview of the overlap between different interesting genomic features, with very little code. An alternative to generate more complex visualisations is the Gviz package from Bioconductor [21]. Gviz creates plots of genomic features and custom data along genomic coordinates in the style of a genome browser. It enables users to query annotations or other information about the genome through online resources such as UCSC or ENSEMBL, and to plot them alongside experimental data. The latter can be displayed in a number of ways (scatter plots, histograms, heatmaps, etc.). The Gviz package is very flexible, enabling the user to construct and customise arbitrarily complex plots. At the same time, it is limited to representing one chromosome (or one part of it) at a time. Therefore, hidecan and Gviz provide complementary ways to visualise different types of genomic features within a single plot.

It is worth mentioning that the goal of hidecan is to generate static summary figures that can be used in reports, presentations, and publications. However, prior to the construction of these summary plots, interactive visualisations can facilitate the exploration of the results, e.g., through the ability to zoom in on specific regions or display additional information through hovering functionalities. This can be achieved by displaying the results in a genome browser, such as the Interactive Genomics Viewer (IGV) [22], the Integrated Genome Browser (IGB) [23], or JBrowse 2 [24]. While this is outside of the scope of the hidecan package, a possible extension of the work would be to enable the generation of BED files that could be used as input for a genome browser from the input data provided by the user to hidecan.

## Figures and Tables

**Figure 1 genes-15-01244-f001:**
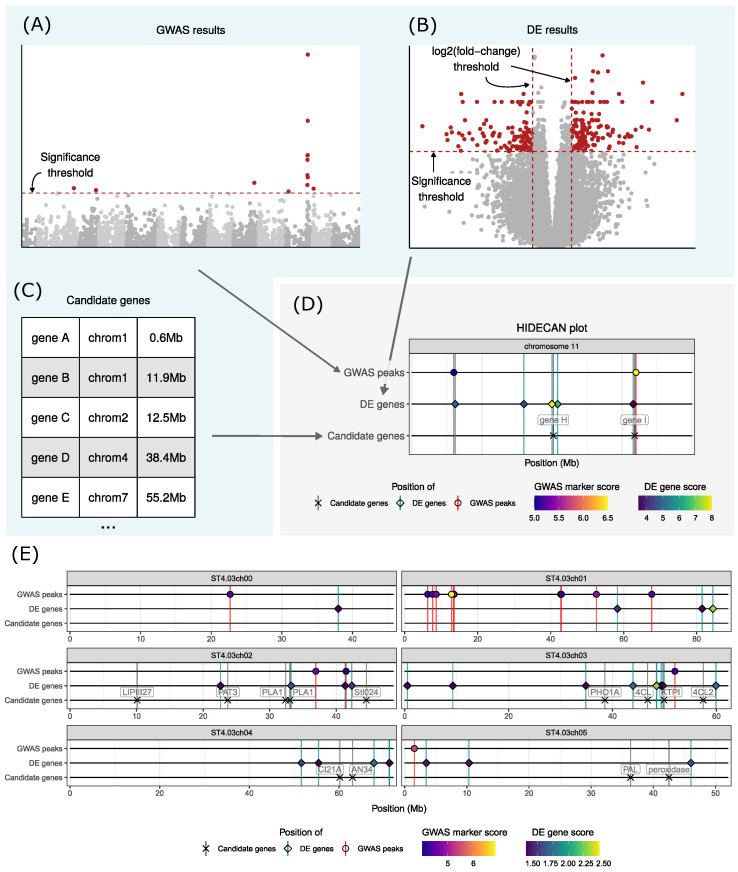
(**A**–**D**): Schema of the construction of a HIDECAN plot. The blue box indicates data and plots obtained outside of the hidecan package. (**A**) From a table of genome-wide association study (GWAS) results (represented here with a Manhattan plot), interesting genomic markers are selected by applying a significance threshold on their association score. (**B**) Similarly, differentially expressed genes are selected from the differential expression (DE) results (represented here with a volcano plot) by applying a threshold on the genes’ scores and log2-fold change. (**C**) Lastly, a manually curated table of candidate genes of interest can be specified. (**D**) The HIDECAN plot displays the genomic position (*x*-axis) of the selected genomic variants, differentially expressed genes, and candidate genes along each chromosome. The colour of the points represents the GWAS and DE scores. In this example, only one chromosome is represented for better clarity. (**E**) A HIDECAN plot generated with the package, from GWAS, DE results, and candidate genes on a potato tuber bruising dataset (only chromosomes 0 to 5 are represented for better clarity; see Appendix A for the full figure). Note that “chromosome 0” is used to group markers or genes that could not be placed on one of the 12 chromosomes.

**Figure 2 genes-15-01244-f002:**
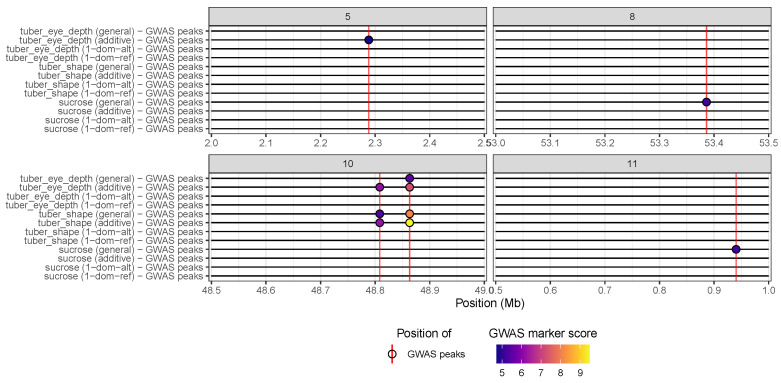
HIDECAN visualisation of GWAS results obtained with the GWASpoly package from an autotetraploid potato dataset. Only chromosomes with significant markers are shown.

**Figure 3 genes-15-01244-f003:**
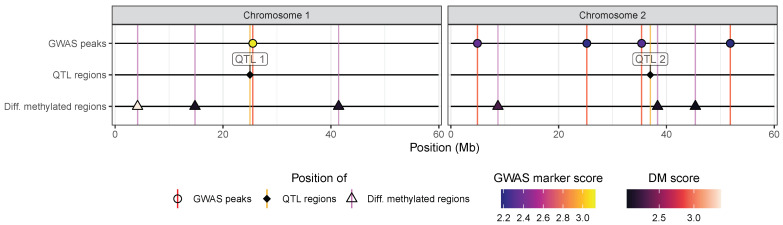
HIDECAN visualisation of simulated GWAS results along with custom user tracks: in this case, QTL regions and differentially methylated regions.

## Data Availability

No new data were generated or analysed in support of this research. The source code for the software is available online at https://github.com/PlantandFoodResearch/hidecan (accessed on 22 September 2024).

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
