# Peer review of "Visual Integration of Genome-Wide Association Studies and Differential Expression Results with the Hidecan R Package"

_genes, 2024, doi:10.3390/genes15101244_

Round 1

Reviewer 1 Report

Comments and Suggestions for Authors

The manuscript by Angelin-Bonnet et al. describes the hidecan R package, which implements a novel visualization of GWAS and differential expression results in a single plot.  The package seems useful and I only have minor comments.

I can imagine applications of HIDECAN plots beyond GWAS and differential gene expression, as mentioned in the last sentence of the discussion.  It seems that the results of assays such as DNA methylation or chromatin accessibility could be plotted in a HIDECAN plot.  Could the hidecan_plot function be further generalized, for example accepting a list of data frames and labels for their respective tracks?  Then we wouldn’t, for example, have the “GWAS peaks” label on results that weren’t necessarily GWAS.

The result displayed in a HIDECAN plot could also be displayed as BED files in a genome browser such as IGV, which could be helpful in exploratory analysis, before generation of a final HIDECAN plot. Visualization in a genome browser may be worth mentioning in the discussion.  Perhaps even a helper function in the hidecan package could generate the BED files, although I understand if that is outside the scope of what the authors are trying to do.

An alternative implementation of HIDECAN might make use of Bioconductor’s GenomicRanges and possibly Gviz packages.  These may be mentioned in the discussion as a means for users to generate more customizable plots.

The shiny app seems very convenient, but I did not find any documentation of the file formats for input.  Should they be tab-delemited?  What column names are required?  Etc.

Reviewer 2 Report

Comments and Suggestions for Authors

Journal: Genes (ISSN 2073-4425)

Manuscript ID: genes-3208002

Type: Article

Title: Visual Integration of GWAS and Differential Expression Results with the hidecan R Package

Section: Plant Genetics and Genomics

Special Issue: Genetics and Genomics of Polyploid Plants

iThenticate

Uploaded at: 2024-08-30 04:09:07

Percent match: 71%

In this research, the authors have developed an R package to analyze and plot the results of GWAS and differential expression studies in an integrated way. I have used the Supplementary Material following the steps indicated by the authors and all was reproducible in a fancy and easy way. I consider this Supplementary Material very useful for any R user. The high percentage in the iThenticate report (71%) is due basically to the preprint publication in an open repository, https://www.biorxiv.org/content/10.1101/2023.03.30.535015v1.full). I have uniquely minor comments/suggestions. The pending issues that need to be addressed are listed below in two sections: (1) Content issues and (2) Formatting issues.

Abstract

—Line 5: Please add a link after the “GWASpoly package”. For instance: https://github.com/jendelman/GWASpoly

Keywords

—Line 12: I strongly recommend adding “R package”, “CRAN”, or something like this as a keyword.

Materials and Methods

—Line 86 (Figure 1 caption): Chromosome 0? It seems a bit strange to me, but maybe it is a lack of knowledge on my part. I have checked different potato chromosome level assemblies at NCBI, and all had chromosomes counting from 1 to 12…Maybe this could be explained in one sentence somewhere.

FORMATTING ISSUES

Below is a non-exhaustive list of typographical or formatting errors. Please check carefully, as there may be more.

Results

—Line 113: Please put Figure 2 below the first paragraph of this section. Figures must be placed after their main text mention.

References

—Line 219: Please change “Solanum Tuberosum L.” to “Solanum Tuberosum L.” (italics).
